# RANet: Region Attention Network for Semantic Segmentation

**Dingguo Shen**[1,2,*], **Yuanfeng Ji**[3,*], **Ping Li**[4], **Yi Wang**[2], **Di Lin**[1,†]

[1]Tianjin University, [2]Shenzhen University, [3]The University of Hong Kong,
[4]The Hong Kong Polytechnic University
{qq237942920, jyuanfeng8}@gmail.com, p.li@polyu.edu.hk
onewang@szu.edu.cn, di.lin@tju.edu.cn

## Abstract

Recent semantic segmentation methods model the relationship between pixels to construct the contextual representations. In this paper, we introduce the *Region Attention Network* (RANet), a novel attention network for modeling the relationship between object regions. RANet divides the image into object regions, where we select the representative information. In contrast to the previous methods, RANet configures the information pathways between the pixels in different regions, enabling the region interaction to exchange the regional context for enhancing all of the pixels in the image. We train the construction of object regions, the selection of the representative regional contents, the configuration of information pathways and the context exchange between pixels, jointly, to improve the segmentation accuracy. We extensively evaluate our method on the challenging segmentation benchmarks, demonstrating that RANet effectively helps to achieve the state-of-the-art results. Code will be available at: `https://github.com/dingguo1996/RANet`.

## 1 Introduction

Recent success of semantic segmentation lies on the deep networks [1, 2, 3] that learn powerful visual representations from the large-scale datasets [4, 5, 6]. The latest segmentation methods [7, 8, 9, 10, 11, 12] model the spatial and category relationship between pixels. They provide rich context for enhancing the representations of pixels and improving the segmentation accuracy.

The spatial pyramid pooling (SPP) and attention mechanism are two of the popular methods, which have been vastly used for constructing the contextual representations. SPP [7, 13, 14, 15] uses various sizes of regular convolutional/pooling kernels to capture the contextual information in different ranges. The attentional models [9, 10, 11, 16] exchange context between each pair of pixels by respecting the dependency between categories. These methods focus on modeling either the spatial or the category relationship between pixels. However, rather than the pixel-level relationship, the semantic segmentation task heavily relies on the understanding of the object-level relationship that provides richer context for classifying the pixels.

In this paper, we advocate the idea of using the object regions for constructing the regional context, which models the object-level relationship. Here, the region refers to an object or an object part. Each object can be regarded as a region that consists of a set of nearby pixels belonging to the same category. Intuitively, the boundaries of the object regions provide the spatial relationship between the objects. In the same object region, the pixels contain the consistent category information. We resort

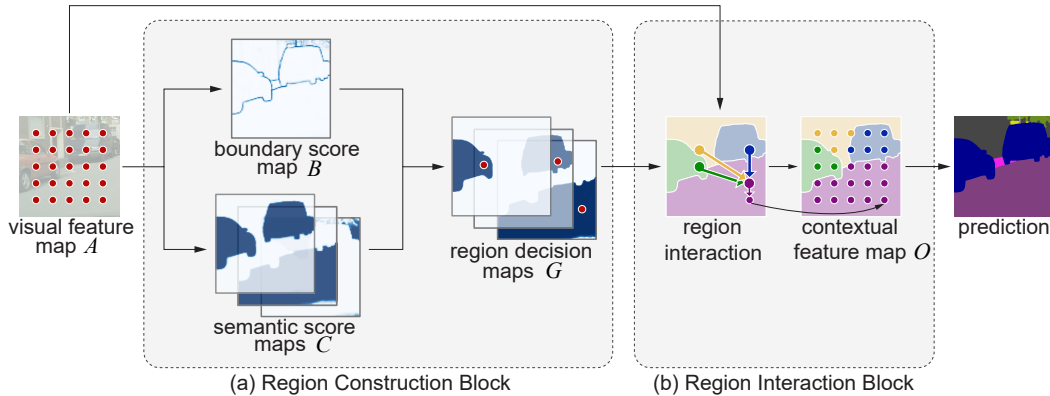

|  (a) Region Construction Block | (b) Region Interaction Block |

Figure 1: RCB (a) constructs a region decision map for each pixel (the red dot), based on the boundary and semantic score maps. RIB (b) selects the representative pixels (the larger dots) in different regions (the yellow, green, blue and purple regions). It enables the region interaction to aggregate the global context, which enhances the pixels (the smaller dots) to produce the contextual feature map. For simplicity, we show only one representative pixel in each object region.

to these nice properties of the object regions and construct the regional context, which enhances the pixel representations and eventually improves the segmentation performance.

Specifically, we propose the *Region Attention Network* (RANet), which consists of the novel network components for constructing the contextual representations. As illustrated in Figure 1(a), RANet employs the *Region Construction Block* (RCB) to jointly analyze the boundary score map and the semantic score maps. It computes the region attention score for each pair of pixels in the image. The high attention score means that the pixels belong to the same object region. Based on the attention scores, RCB computes the region decision map for each pixel (red dot). It uses the region decision maps for dividing the image into different object regions.

Next, the region decision maps are passed to the *Region Interaction Block* (RIB). As illustrated in Figure 1(b), RIB selects the representative pixels (the larger dots) in different regions. Within the same region, each representative pixel receives the context from other pixels. It allows the representative pixels to effectively represent the local content of the object region. Moreover, RIB communicates the representative pixels in different regions, comprehensively capturing the spatial and category relationship between different object regions. RIB yields the global contextual representation to augment the pixels (the smaller dots), finally forming the contextual feature map for segmentation.

We evaluate RANet on the challenging segmentation datasets. We achieve 84.0, 54.9 and 40.7 mean Intersection-over-Unions (IoUs) on the Cityscape test set [4], PASCAL Context validation set [5] and COCO-Stuff validation set [6], respectively. These results demonstrate the effectiveness of all of the network components (i.e., RCB and RIB), which improve the segmentation accuracy and help to surpass the state-of-the-art methods.

## 2 Related work

The literature on image segmentation [7, 13, 17, 18, 9, 12] has demonstrated the effectiveness of the *Multi-Scale Context Aggregation* and the *Attention Mechanism*, which are closely related to our work in the sense that they fuse the pixels to form the contextual representations for segmentation.

### 2.1 Multi-Scale Context Aggregation

There has been tremendous progress on the semantic segmentation task, thanks to the fully convolutional network (FCN) [1] that is good at modeling the pixel-wise visual information. Recently, an array of research works aggregate multiple pixels for achieving the multi-scale contextual information. Typically, SPP [7, 13, 19] uses the different sizes of convolutional/pooling kernels to fuse the visual representations of pixels, forming the contextual representations. The Encoder-Decoder networks [14, 15, 20, 21] merge the convolutional feature maps at different network layers. Moreover, GCN [22], EncNet [18] and ParseNet [23] utilize the relatively large sizes of convolutional/pooling

kernels to harness more global context. These methods leverage the image contents in different sizes of receptive fields to compute the multi-scale contextual representations, but neglecting the shapes of the object regions that are critical to determining the object categories in different ranges.

In this paper, we use the category and the boundary information to construct the contextual representations. We use the pixels in the same object region to construct the local context to represent the regional content. Furthermore, our approach enables the exchange of the regional information to yield the global context, which models the relationship between object regions. The latest works [12, 24] also use the boundary information to discriminate the pixels in different object regions. They may produce the discrepant representations for the pixels, which are far away from each other but belong to the same category. In contrast, we follow the non-local attention [17] to directly communicate the pixels in different object regions and produce more appropriate visual representations.

## 2.2 Attention Mechanism

Attention mechanism has been used for constructing the contextual representations. Chen et al. [25] use the attention mask to model the dependency between pixels in different ranges. Hu et al. [26] use the global pooling to achieve a wider range of context. The latest non-local attention methods [9, 10, 17, 27] connect each pair of pixels to precisely model their relationship. Wang et al. [28] inject the position information of pixels into the context. Fu et al. [16] incorporate the feature-channel attention with the position attention [9, 10, 17, 27] to model the semantic and spatial relationship between pixels in a more comprehensively manner. However, the pixel-wise attention methods may construct the contextual representation based on the pixels, which are misclassified by the deep network. The problematic context easily leads to the misclassification of the highly-correlated pixels.

Different from the pixel-wise attention mechanism, we use the region attention to exchange the regional context between pixels. We select the representative pixels to represent each object region. We use the representative pixels to distill the context and form the consistent regional information. The representative pixels in different regions exchange regional information to enrich the context.

## 3 Method

We use the regional context, which models the spatial and category dependency between pixels, to assist the image segmentation. The is done by the step-by-step RCB and RIB, to satisfy the need for using the regional context to enhance the pixels. We use RCB to group the pixels of the image reasonably into different object regions. Based on the boundary and the representative pixels of the region, we construct the spatial and category representations of the object. Next, RIB exchanges information between object regions, forming the regional context for enhancing the pixels.

Our approach is schematically illustrated in Figure 1. We use the boundary score map and the semantic score maps to compute an attention score for each pair of pixels. In Figure 1(a), RCB recognizes the high attention score and groups the corresponding pair of pixels to the same region, yielding the the region decision maps $F \in \mathbb{R}^{(H \times W) \times (H \times W)}$. Each pixel (the red dot) is associated with an region decision map to find the pixels belonging to the same region. RCB yields $Q$ object regions $\{R_q | q = 1, \ldots Q\}$. Different images may have different numbers of object regions.

In Figure 1(b), we compute the representative score for each pixel, based on its correlation with other pixels in the same region. In the $q^{th}$ object region, RIB selects $K$ representative pixels (the larger dots) $\{p_{q,k} | p_{q,k} \in \phi(R_q), k = 1, ..., K\}$, which have the highest representative scores in $R_q$. Given the representative pixels, RIB enables the region interaction to enhance each pixel (the smaller dot) in the image. At first, RIB conducts the *intra-region collection*, where the representative pixel $p_{q,k}$ receives the information from the pixels in $R_q$. Next, RIB uses the *inter-region interaction* for propagating the regional information of $p_{q,k}$ to the representative pixels in other regions, while $p_{q,k}$ also receives the regional information. Finally, RIB uses the *intra-region distribution* to propagate the augmented information of $p_{q,k}$ to all of the pixels in $R_q$. Each pixel receives the information from $K$ representative pixels in the same region. By using the regional information to enhance all of the pixels in the image, we yield the contextual feature map $O \in \mathbb{R}^{H \times W \times M}$.

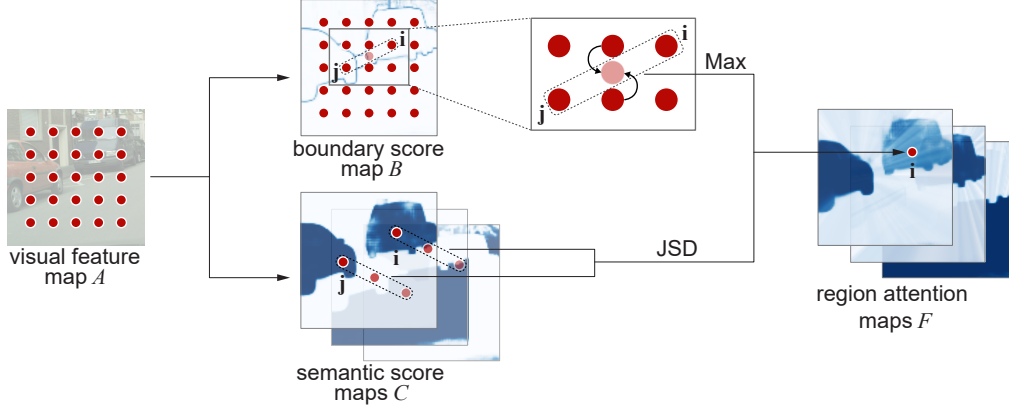

Figure 2: RCB provides the region attention map for each pixel (the red dot) in the image. In the boundary score map, the dot in lighted red is associated with the boundary score, which is computed by the linear interpolation of the boundary scores of the two nearest pixels.

## 3.1 Region Construction Block

In Figure 2, we learn the visual feature map $A \in \mathbb{R}^{H \times W \times M}$ from the image. $H \times W$ denotes the resolution. $M$ denotes the number of channels. $A$ is passed to two separate convolutional layers to achieve the boundary score map $B \in \mathbb{R}^{H \times W}$ and the semantic score maps $C \in \mathbb{R}^{H \times W \times N}$. $N$ denotes the number of object categories. In $B$, each pixel has a boundary score for locating the object boundary. In $C$, each pixel is associated with the probabilities of $N$ categories. We use $B$ and $C$ to group the pixels to form different object regions. In a region, each pair of pixels has a low probability of having the in-between boundary and a high probability of belonging to the same category.

We compute the probability $D_{i,j} \in [0,1]$ of finding the boundary between the $i^{th}$ and $j^{th}$ pixels as:

$$D_{i,j} = \max(B_{i \leftrightarrow j}), \tag{1}$$

where $B_{i \leftrightarrow j}$ denotes a set of pixels on the line, whose end points are the the $i^{th}$ and $j^{th}$ pixels (see the upper branch of RCB in Figure 2). In $B_{i \leftrightarrow j}$, each pixel has a boundary probability. A smaller value of $D_{i,j}$ means a lower probability of finding the boundary between the $i^{th}$ and $j^{th}$ pixels.

Next, we use the differentiable and symmetrical *Jensen-Shannon Divergency* (JSD) to compute the semantic similarity $E_{i,j} \in [0,1]$ between the $i^{th}$ and $j^{th}$ pixels as:

$$E_{i,j} = \sum_{n=1}^{N} \frac{C_{i,n} \log C_{i,n} + C_{j,n} \log C_{j,n}}{2U_{i,j,n}}, \quad U_{i,j,n} = \frac{C_{i,n} + C_{j,n}}{2}. \tag{2}$$

$C_i, C_j \in \mathbb{R}^N$ are the vectors of category probabilities for the $i^{th}$ and $j^{th}$ pixels. A smaller value of $E_{i,j}$ means a higher probability of identifying the $i^{th}$ and $j^{th}$ pixels as the same category.

Finally, we generate the region attention maps $F \in \mathbb{R}^{(H \times W) \times (H \times W)}$ as:

$$F_{i,j} = (1 - D_{i,j})(1 - E_{i,j}), \tag{3}$$

where $F_{i,j}$ is an attention score. A larger value of $F_{i,j}$ means higher probability of grouping the $i^{th}$ and $j^{th}$ pixels to the same object region. We model the grouping process (see Figure 3) as:

$$G_{i,j} = \frac{1}{2} sgn(\frac{F_{i,j}^g + F_{j,i}^g}{2}) + \frac{1}{2}, \quad F^g = W^g \otimes F, \tag{4}$$

where $sgn$ denotes the sign function, whose output is $-1$ or $1$. $G \in \mathbb{R}^{(H \times W) \times (H \times W)}$ denotes the symmetrical region decision maps, where $G_{i,j} = G_{j,i}$ and $G_{i,j} \in \{0,1\}$. $W^g$ denotes the convolutional kernel. We learn richer information from the region attention maps $F$ to form the regularized attention score maps $F^g \in \mathbb{R}^{(H \times W) \times (H \times W)}$. $G_{i,j} = 1$ indicates that the $i^{th}$ and $j^{th}$ pixels are grouped to the same region; otherwise, the pixels belong to different regions. The grouping process divides the image into a set of regions $\{R_q | q = 1, \ldots Q\}$.

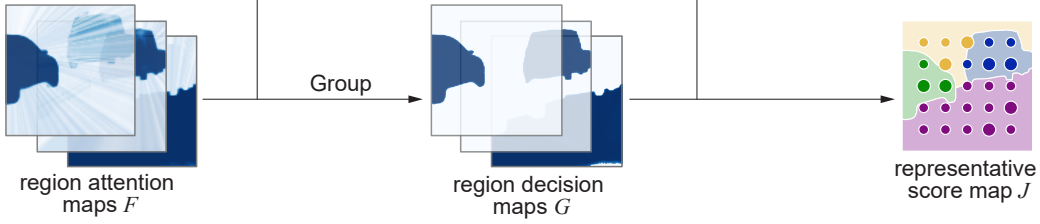

Figure 3: RCB uses the region attention maps to construct the region decision maps and the representative score map. In the representative score map, the larger dots in different colors represent the representative pixels in different regions.

## 3.2 Region Interaction Block

As illustrated in Figure 1(b), we pass the region decision maps $G$ to RIB, which selects the representative pixels of object regions to enable the region interaction. For the $i^{th}$ pixel of the $q^{th}$ region, we compute the representative score $J_i$ as:

$$J_i = \frac{1}{|R_q|} \sum_{j \in R_q} F_{i,j}^g, \tag{5}$$

where $J_i \in [0, 1]$ represents the correlation between the $i^{th}$ pixel and other pixels in the $q^{th}$ region. We compute the representative scores for all of the pixels to produce the representative score map $J \in \mathbb{R}^{H \times W}$. We select $K$ representative pixels $\{p_{q,k} | p_{q,k} \in \phi(R_q), k = 1, ..., K\}$, which have the highest representative scores among all of the pixels in the $q^{th}$ region. The set $\phi(R_q)$ contains the locations of the representative pixels in the $q^{th}$ region. Note that the region decision maps $G$ and the representative score map $J$ contain the category information of object regions. They also contain the the boundary information for modeling the spatial relationship between regions. They are used by RIB for injecting the regional context into the contextual representations.

The representative pixels provide the regional information, which is propagated between the pixels in different regions. Figure 4 provides more details. Firstly, RIB conducts the intra-region collection to construct the local contextual representations for all of the representative pixels. As illustrated in Figure 4(a), the representative pixels (the larger dots in yellow) receive the information from the pixels (the smaller dots) in the same region. For the $k^{th}$ representative pixel of the $q^{th}$ region, the intra-region collection computes the local contextual representation $A_{q,k}^l \in \mathbb{R}^M$ as:

$$A_{q,k}^l = J(p_{q,k})A(p_{q,k}) + \sum_{i \in R_q \setminus \phi(R_q)} W_{q,k,i}^l(J_i A_i), \quad W_{q,k,i}^l = \frac{\exp\left(A(p_{q,k})(J_i A_i)^\top\right)}{\sum_{j \in R_q \setminus \phi(R_q)} \exp\left(A(p_{q,k})(J_j A_j)^\top\right)}, \tag{6}$$

where $A(p_{q,k}) \in \mathbb{R}^M$ and $J(p_{q,k}) \in \mathbb{R}$ denote the visual feature and the representative score of the representative pixel $p_{q,k}$, respectively. $A_i, A_j \in \mathbb{R}^M$ denote the visual features of the $i^{th}$ and $j^{th}$ pixels in the region $R_q$. These visual features are extracted from the visual map $A$.

The local contextual representations of all of the representative pixels are used by the inter-region interaction, as illustrated in Figure 4(b). We build the shortcut connection between each pair of the representative pixels in different regions. For the representative pixel $p_{q,k}$, we aggregate the representative pixels in other regions, forming the global contextual representation $A_{q,k}^g \in \mathbb{R}^M$ as:

$$A_{q,k}^g = J(p_{q,k})A_{q,k}^l + \sum_{s=1, s \neq q, i \in \phi(R_s)}^{Q} W_{q,s,k,i}^g(J(p_{s,i})A_{s,i}^l),$$

$$W_{q,s,k,i}^g = \frac{\exp\left(A_{q,k}^l(J(p_{s,i})A_{s,i}^l)^\top\right)}{\sum_{p_{s,j} \in \phi(R_s)} \exp\left(A_{q,k}^l(J(p_{s,j})A_{s,j}^l)^\top\right)}. \tag{7}$$

We use the weight $W_{q,s,k,i}^g$ and the representative score $J(p_{s,i})$ to control the spatial and category information, which is propagated from the $i^{th}$ representative pixel in the $s^{th}$ ($s \neq q$) region to the $k^{th}$ representative pixel in the $q^{th}$ region.

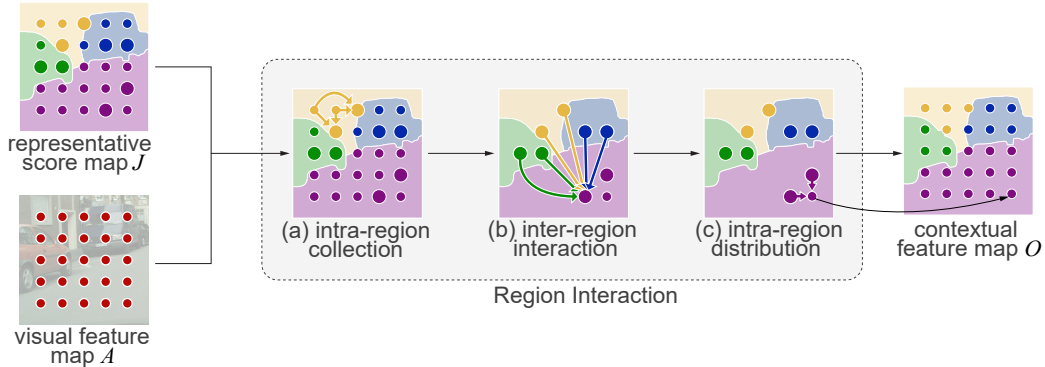

Figure 4: RIB uses the intra-region collection (a), the inter-region interaction (b) and the intra-distribution (c) to propagate the regional information to the pixels in differen regions.

Finally, RIB uses the intra-region distribution (see Figure 4(c)) to propagate the global context for augmenting all of the pixels in the image. For the $i^{th}$ pixel in the $q^{th}$ region, we compute the contextual representation $O_i \in \mathbb{R}^M$ as:

$$O_i = J_i A_i + \sum_{p_{q,k} \in \phi(R_q)} W_{q,k,i}^d (J(p_{q,k}) A_{q,k}^g), \quad W_{q,k,i}^d = \frac{\exp\left(A_i (J(p_{q,k}) A_{q,k}^g)^\top\right)}{\sum_{p_{q,j} \in \phi(R_q)} \exp\left(A_i (J(p_{q,j}) A_{q,j}^g)^\top\right)}. \tag{8}$$

We compute the contextual representations for all of the pixels and form the contextual representation map $O \in \mathbb{R}^{H \times W \times M}$, which is used for segmenting the image.

# 4   Implementation Details

We construct RANet with the Pytorch toolkit. We use the 8-stride ResNet-101 [29] pre-trained on the ImageNet [30] as the backbone, where the *res5* layer provides the visual feature map for RANet. The ground-truth segmentation mask is used for supervising the prediction of the semantic score maps in RCB and the final segmentation mask. Based on the ground-truth segmentation mask, we follow [31, 32] to compute the binary boundary mask for supervising the prediction of the boundary score map in RCB. We use the cross-entropy loss to penalize all of the training error.

We augment the training images with the horizontal flipping, random brightness jittering and scaling. We use the standard SGD solver with the initial learning rate of 0.001 to train the network. The learning rate is decayed linearly during the training. Each mini-batch contains 8 images. For the Cityscape dataset, we use the image size of $769 \times 769$ and 40,000 mini-batches to train the network. For the PASCAL-Context and COCO-Stuff datasets, we set the image to $520 \times 520$ and use 60,000 mini-batches. We train the network on 8 TITAN XP. Given the trained network, we employ the different scaling factors (i.e., 0.5, 0.75, 1.0, 1.25, 1.5, 1.75) to achieve the multi-scale testing result.

# 5   Experimental Results

We conduct the experiments on the Cityscapes, PASCAL Context and COCO-Stuff datasets. The Cityscapes dataset contains 2,975 training, 500 validation, and 1,525 testing images, with the annotations of 30 categories in the urban scene. We focus on the 19 challenging categories. The PASCAL Context dataset contains 59 categories and background, providing 4,998 training and 5,105 testing images. The COCO-Stuff dataset provides 9,000 training and 1,000 validation images, with the rich annotations of 171 categories (i.e., 80 object classes and 91 stuff classes). Below, we mainly evaluate our approach on the Cityscapes validation set to show the effectiveness of all of the network components. Finally, we compare our approach to the state-of-the-art methods on all of the datasets. We report the segmentation result in terms of the mean intersection-over-union (IoU).

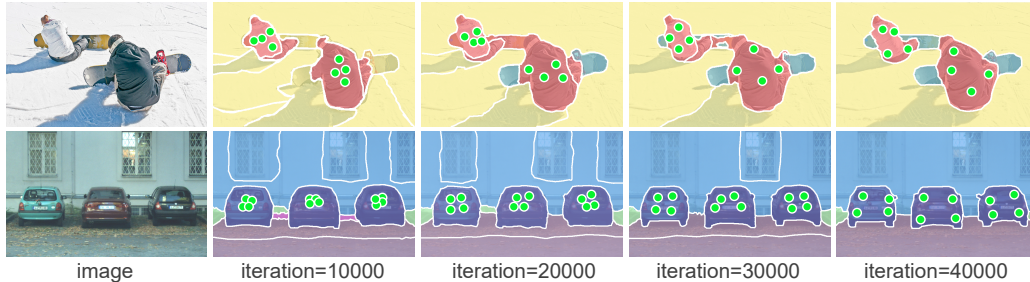

| image | iteration=10000 | iteration=20000 | iteration=30000 | iteration=40000 |

Figure 5: The representative pixels (the green dots) and the object regions in different iterations of the the network training. To simplify the visualization, we show only one category of the regions in an image. Each region has 4 representative pixels .

## 5.1  Analysis of the Network Components

We examine the importance of the core network components of RANet to the segmentation accuracy. Compared to the backbone network that achieves 77.5% mean IoU on the Cityscapes validation set, RANet uses the powerful RCB and RIB to achieve the accuracy of 81.9%.

| method | pixel-wise context | | region-wise context | | |
|---|---|---|---|---|---|
| | SPP | non-local attention | boundary map | semantic map | w/ both |
| mean IoU | 79.6 | 80.5 | 80.8 | 81.3 | **81.9** |

Table 1: Comparison of different methods for propagating context. The performance is evaluated on the Cityscapes validation set. The segmentation accuracy is reported in terms of mean IoU (%).

In Table 1, we remove RCB to disable the exchange of the regional context. We use SPP [7] and the non-local attention model [17] to enable the local and global pixel-wise context propagation, respectively, yielding lower accuracies than our region attention model. Note that these methods communicate the pixels without using the regional information to enrich the context. We further investigate different methods for constructing the object regions for the context exchange. We use the boundary score map or the semantic score map in an isolated manner (see the entries "boundary map" and "semantic map"), to predict the region attention map. Compared to RANet, the independent boundary map (or the semantic map) leads to a lower accuracy. It demonstrates the effectiveness of combining the spatial and category information to model the relationship between object regions.

RIB uses the regional information to form richer context. We disable the intra-region communication (i.e., the intra-region collection and distribution) and exchange context between the representative pixels in different regions. This strategy only updates the the representative pixels. We also investigate the case of removing the inter-region interaction. Here, we only propagate context between the pixels within the same region. These strategies degrade the performance to 79.2% and 81.1% mean IoUs.

## 5.2  Sensitivity to the Representative Pixels

We change the number of the representative pixels and study the effect on the segmentation accuracy. There are 400 pixels in each region, on average. In Table 2, we follow [11] to select 1% of the pixels (i.e., $K = 4$) from each region, yielding the accuracy of 81.6%. Using less representative pixels degrades the performance, because the important regional content may be lost. But the redundant representative pixels may include the pixels near the object boundary, adding the ambiguous information to context. We set $K = 8$ to improve the performance reasonably. We show the examples of the representative pixels and the object regions in Figure 5. The representative pixels spread on the object regions, capturing the useful regional information to refine the segmentation result gradually.

We provide more details of the computational overheads (i.e., the floating-point operations per second (FLOPS) and the GPU memory), which are required by RANet. We compare RANet with the latest pixel-wise attention models [11, 10, 16, 9] in Table 3, where RANet achieves a better performance at the cost of the reasonable computational overheads. Note that the asymmetric non-local model (Asymmetric NL) [11] also propagates the information from the representative pixels to other pixels,

| K | 2 | 4 | 6 | 8 | 10 | 12 |
|---|---|---|---|---|---|---|
| mean IoU | 81.2 | 81.6 | 81.7 | **81.9** | 81.8 | 81.6 |

Table 2: Sensitivity to the number of the representative pixels. The performance is evaluated on the Cityscapes validation set. The segmentation accuracy is reported in terms of mean IoU (%).

| method | ▲ FLOPS (G) | ▲ parameters (M) | ▲ memory (MB) | mIoU |
|---|---|---|---|---|
| Asymmetric NL[11] | **163** | **13** | **634** | 81.0 |
| CCNet[10] | 198 | 18 | 730 | 81.1 |
| DANet[16] | 264 | 23 | 2611 | 81.5 |
| PSANet[9] | 372 | 50 | 3124 | 80.6 |
| RANet | 212 | 21 | 1893 | **81.9** |

Table 3: Comparisons with other attention methods. ▲ denotes increase in FLOPS/the number of parameters/GPU memory, which is estimated by using the $[1 \times 512 \times 97 \times 97]$ feature map.

thus saving computation. It selects the representative pixels, which have the high activation values in the high-dimensional feature space. RANet uses the explicit boundary and semantic scores to select the representative pixels, outperforming Asymmetric NL by about 1% mean IoU.

### 5.3 Comparison of the Coarse and Fine Segmentation Results

Note that the semantic score map, which is used by RCB to compute the region attention map, can be regarded as the coarse segmentation result of the input image. In Table 4, we compare the coarse segmentation results with the fine segmentation results, which are achieved by the full RANet, on different datasets. Because the full model is equipped with RCB and RIB for computing the powerful contextual representations, the fine segmentation lead to better segmentation accuracies than the coarse segmentation.

| method | Cityscapes | PASCAL Context | COCO |
|---|---|---|---|
| coarse segmentation | 77.3 | 51.2 | 37.9 |
| fine segmentation | **81.9** | **54.9** | **40.7** |

Table 4: Comparison of the coarse and fine segmentation results on the Cityscapes, PASCAL Context and COCO-Stuff validation sets. The segmentation accuracy is reported in terms of mean IoU (%).

### 5.4 Comparisons with State-of-the-art Methods

In Table 5, we compare RANet with state-of-the-art methods on the Cityscapes test set. We use the fine annotations to train RANet and achieve the accuracy of 82.4% mean IoU, which is comparable to the latest methods based on more powerful baseline networks [33, 34]. Furthermore, we use the fine annotations along with the extra coarse annotations to train RANet, which is based on a stronger baseline model [35] (i.e., "HRNetV2-W48+ASPP" in Table 5), yielding a better accuracy of 84.0%. In Table 6, we report the segmentation accuracies on the PASCAL Context and COCO-Stuff validation sets. For a fair comparison, we compare RANet with the latest methods that also use ResNet-101 as the backbone model. Again, we achieve better accuracies than other methods. We provides the examples of the segmentation results in Figure 6.

## 6 Conclusions

We have proposed a novel region attention network for modeling the dependency between the object regions to compute the contextual representations. Our approach learns the object regions based on the boundary and category information. We select the representative pixels from the object regions and construct the regional context for the intra-region and inter-region communications between pixels. Our approach is effective and outperforms the state-of-the-art on several public datasets.

| method | | backbone | annotation | |
|---|---|---|---|---|
| | | | w/o coarse | w/ coarse |
| SPP | PSPNet[7] | ResNet-101 | 80.1 | 81.2 |
| | Deeplabv3+[36] | Xception-71 | 81.0 | 81.9 |
| | DPC[33] | Xception-71 | 82.7 | - |
| attention | Asymmetric NL[11] | ResNet-101 | 81.3 | - |
| | CCNet[10] | ResNet-101 | 81.4 | - |
| | OCRNet[34] | HRNetV2-W48+ASPP | 83.2 | 83.7 |
| | RANet | ResNet-101 | 82.4 | 83.0 |
| | | HRNetV2-W48+ASPP | **83.4** | **84.0** |

Table 5: Comparisons with other state-of-the-art methods on the Cityscapes test set.

| | PASCAL Context | | COCO-Stuff | |
|---|---|---|---|---|
| | method | mIoU | method | mIoU |
| SPP | SVCNet[24] | 53.2 | DSSPN[37] | 38.9 |
| | HRNet[35] | 54.0 | SVCNet[24] | 39.6 |
| attention | BFP[12] | 53.6 | CCNet[10] | 39.8 |
| | CFNet[38] | 54.0 | EMANet[39] | 39.9 |
| | ACNet[40] | 54.1 | ACNet[40] | 40.1 |
| | RANet | **54.9** | RANet | **40.7** |

Table 6: The segmentation accuracies on the PASCAL Context and COCO-Stuff validation sets.

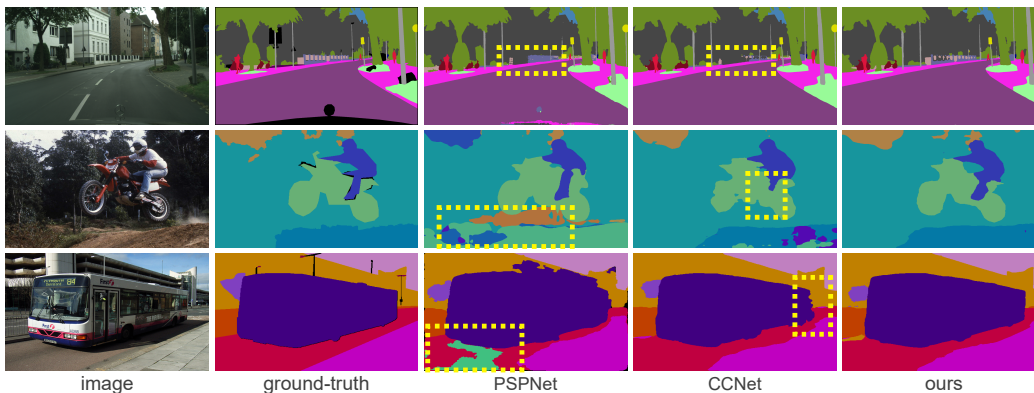

Figure 6: Segmentation results on the Cityscapes, PASCAL Context and COCO-Stuff validation sets.

# Broader Impact

Our approach can help to achieve rich scene information, based on the large-scale image data that have been capturing by the cameras. It boosts the development of the AI systems (e.g., autonomous vehicle and video surveillance) in many scenarios. One should be cautious of using the data source, which belongs to the official or private organization, for training our segmentation model. This may give rise to the infringement of privacy or economic interest. The problematic segmentation results may lead to the misleading information, which may be released to the public.

# Acknowledgments

We thank the anonymous reviewers and editors for their constructive suggestions. This work was supported in parts by NSFC (61702338, 61701312), Natural Science Foundation of Guangdong Province (2019A1515010847), the Macau Science and Technology Development Fund under grant (0027/2018/A1), and The Hong Kong Polytechnic University under grants (P0030419, P0030929).

## Footnotes

*The first two authors share the contribution equally.

†Di Lin is the corresponding author of this paper.

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
