[Supplementary Material]

# RANet: Region Attention Network for Semantic Segmentation
# – Supplementary Material –

**Dingguo Shen**[1,2,*], **Yuanfeng Ji**[3,*], **Ping Li**[4], **Yi Wang**[2], **Di Lin**[1,†]
[1]Tianjin University, [2]Shenzhen University, [3]The University of Hong Kong,
[4]The Hong Kong Polytechnic University
{qq237942920, jyuanfeng8}@gmail.com, p.li@polyu.edu.hk
onewang@szu.edu.cn, di.lin@tju.edu.cn

## 1   Training Objective

There are three cross-entropy losses for supervising the learning of the semantic score map, the boundary score maps and the final segmentation mask. Thus, we formulate the training objective $L$ as:

$$L = \lambda_c \cdot L_c + \lambda_b \cdot L_b + \lambda_m \cdot L_m, \qquad (1)$$

where $L_c$, $L_s$, and $L_m$ represent the loss for the semantic score map, boundary score maps and final segmentation mask, respectively. Here, $\lambda_c$, $\lambda_b$, and $\lambda_m$ are used for weighting the losses. Empirically, we set $\lambda_c = 0.4$, $\lambda_b = 0.04$ and $\lambda_m = 1$.

Figure 1: Comparison of the coarse and fine segmentation results on the Cityscapes, PASCAL Context and COCO-Stuff validation sets.

---

[*]The first two authors share the contribution equally.

[†]Di Lin is the corresponding author of this paper.

## 2 Different Strategies of Constructing the Object Regions

In RCB, we compute the probability $D_{i,j}$, which is used for predicting the boundary between the $i^{th}$ and $j^{th}$ pixels, by finding the maximum boundary score of the pixels on the line (see the dash line in Figure 2) connecting the $i^{th}$ and $j^{th}$ pixels. Note that the object region is unnecessarily the convex shape. As illustrated in Figure 2, even the $i^{th}$ and $j^{th}$ pixels belong to the same object region, they lead to a high value of $D_{i,j}$. To alleviate the problem of assigning the $i^{th}$ and $j^{th}$ pixels to different regions, we consider the semantic similarity between the $i^{th}$ and $j^{th}$ pixels when constructing the object regions.

Another trivial strategy (see Figure 2) for computing the boundary probability is to find an intermediate pixel $m'$ as:

$$m' = \arg\min_m (\frac{D_{i,m} + D_{m,j}}{2}), \ \ s.t., m \neq i, m \neq j. \tag{2}$$

We use the boundary probability $D'_{i,j} = \frac{D_{i,m'} + D_{m',j}}{2}$ in place of $D_{i,j}$ to construct the object regions. Based on Eq. (2), we can use dynamic programming to find two or more intermediate pixels for computing the boundary probability, which is more reliable in terms of dealing with the complex object shapes. However, using the intermediate pixels requires extra computation. In Table 1, we compare the strategies with/without using the intermediate pixel. We find that using an intermediate pixel for each pair of pixels slightly improves the segmentation accuracies (0.1~0.2 mean IoU) on different datasets, at the cost of 113G FLOPS and 419MB memory. In Figure 3, we provide the segmentation results with/without using the intermediate pixel. To achieve the satisfactory segmentation result at the cost of the reasonable computation, we choose the strategy without using any intermediate pixel.

Figure 2: The illustration of different strategies of computing the boundary probability.

| method | ▲ FLOPS (G) | ▲ memory (MB) | CS | PC | COCO |
|---|---|---|---|---|---|
| w/o intermediate pixel | **212** | **1893** | 81.9 | **54.9** | 40.7 |
| w/ intermediate pixel | 325 | 2312 | **82.0** | **54.9** | **40.9** |

Table 1: Comparison of different strategies of constructing the object regions. ▲ denotes increase in FLOPS/GPU memory, which is estimated by using the $[1 \times 512 \times 97 \times 97]$ feature map. CS, PC and COCO denote the Cityscapes, PASCAL Context and COCO-Stuff validation sets. The segmentation accuracy is reported in terms of mean IoU (%).

## 3 Different Strategies of Using the Representative Scores

In Table 2, we compare different strategies of using the representative scores in the region interaction. By removing the representative scores, the region interaction only relies on the non-local attention scores. We also study the strategy of using only the representative scores in the region interaction. These two strategies yields lower performances than our full model, which uses the non-local attention scores and the representative scores to control the contextual information exchanged between pixels.

image      w/o intermediate pixel      w/ intermediate pixel

Figure 3: Different strategies of using the intermediate pixels make little difference to the segmentation results. The images are taken from the Cityscapes, PASCAL Context and COCO-Stuff validation sets.

| method | non-local attention score | representative score | w/ both |
|---|---|---|---|
| mean IoU | 80.8 | 81.4 | **81.9** |

Table 2: Comparison of different strategies of using the representative scores on the Cityscapes validation set. The segmentation accuracy is reported in terms of mean IoU(%)

.