[Reviews · NeurIPS 2020]

Review 1

Summary and Contributions: The manuscript proposes an attention network for modeling the relationship between object regions and finally contributes to semantic segmentation. The author proves such an attention mechanism contributes to the contextual representation.

Strengths: The operation in the proposed method is closely combined with mathematics, and it is not a random fabrication. 2) Experiments show that the method proposed by the author is capable and brings useful improvement on multiple datasets. The effect of this method is excellent compared with SPP based and attention-based methods.

Weaknesses: However, there are some concerns to be further improved as well: 1) The author gives the operation of RCB and RIB step by step but does not give a reasonable starting point. Although the final experiment has proved that such an operation may be effective, the author needs a more explicit motivation in this paper. 2) The contrast experiment compares the backbone network's effect, but the backbone network cannot be directly used for segmentation, so the author adds what structure to complete the segmentation after the backbone network, and the subsequent structure will have a significant impact on the segmentation results. The author should explain clearly. 3) There is a large amount of accumulation in the typesetting of the formula in this paper, and the author can further optimize and adjust it.

Correctness: Yes.

Clarity: Yes.

Relation to Prior Work: Yes.

Reproducibility: Yes

Additional Feedback: After the rebuttal: The authors respond to my concerns well. So I update my score with "7: accept".


Review 2

Summary and Contributions: This paper introduces an Region Attention Network (RANet) to model the relationship between object regions for image semantic segmentation; An Region Construction Block (RCB) is designed to jointly analyze the boundary score map and the semantic score maps; An Region Interaction Block (RIB) is designed to select the representative pixels in each region for context information exchanging.

Strengths: The proposed blocks are somewhat novel and the experimental results are good. Ablation analysis are conducted to validate the efficacy of the proposed modules and the whole network.

Weaknesses: The motivation and illustration are not clear. The details are as follows, 1. Why the RANet can capture more context information than SPP and previous attentional models? The authors claim that RANet naturally provides the spatial and category relationship of pixels to construct the contextual representations, but the category information in RANet may be not accurate, which could results in error guiding information. 2. After obtaining the contextual representation $O$ as described by Eq.(8), how to get the final segmentation map? 3.In Eq.(1), $B_{i \arrow j}$ denote a set of pixels on the line, what is the direction of the line? Vertical?horizontal?or oblique?

Correctness: correct

Clarity: The written and illustration should be improved.

Relation to Prior Work: Could be improved, especially the motivation of the RANet, i.e., why it can capture more context information than SPP and previous attentional models?

Reproducibility: No

Additional Feedback: 1. I would like to ask the authors to clarify the motivation of the RANet, i.e., why it can capture more context information than SPP and previous attentional models? 2. How to get the final segmentation map by $O$ obtained by Eq.(8)? 3. In Eq.(1), $B_{i \arrow j}$ denote a set of pixels on the line, what is the direction of the line?


Review 3

Summary and Contributions: The paper introduces a novel attention network for semantic segmentation task. There are two main steps, which is dividing the image into regions and modeling the relation between regions. The author thinks the SPP based methods just use the spatial information between pixels to capture contextual information while other attention methods just use the category information between pixels to exchange context. Compared to these above methods, the new network can better use the correlation between the spatial and category information to enrich the contextual information by exchanging the regional information. Besides, many detailed experiments in various datasets has shown the performance improvement.

Strengths: —It is a novel idea to use the object regions to construct the contextual representations by region interaction —The region construction block and the region interaction block are clearly explained and the visualization is also very good — Extensive experiments are conducted and results are state of the art.

Weaknesses: — The RCB block and RIB block both seem to be very time-consuming. The RCB block includes the boundary score map and category score map. Each map needs to calculate the similarity between each pair of pixels. The RIB block needs to find the representative pixels for each region and aggregate information between regions. So during the training process, it needs to compute all the above information in each iteration, thus it may seriously affect the training speed. —In line 18, authors are encouraged to give more intuitive descriptions to show the differences between RAN and SPP or other attention mechanisms. This may be the main motivation of the new network. —In line 135, In the RIB block, why should we choose the representative pixels? why not use all the pixels in each region?The author needs to give a short explanation. I guess that maybe there are two reasons. The first one is the computation efficiency and the other one is the segmentation accuracy, which is illustrated in table2.

Correctness: YES

Clarity: YES

Relation to Prior Work: YES

Reproducibility: No

Additional Feedback: —In Figure 5, why do these representative pixels gradually separate from each other as the iteration progresses? —Update:The authors have answered my questions well, so I changed my score. I suggest that the author update the motivation of this method in the paper.

[Author Response · NeurIPS 2020]

We sincerely thank the reviewers for their comments. We are pleased to see that our contribution is unanimously found novel the reviewers. The reviewers believe that our experimental results are adequate and convincing. *R1* and *R3* find that our technical details are clearly explained. We will release the source code and model for reproducibility.

**Motivation of our approach:** The popular SPP and attention approaches construct the contextual representation that captures the visual relationship between pixels. However, the semantic segmentation task heavily relies on the understanding of the object-level relationship that is ineffectively captured by the pixel-level context. It motivates us to leverage the object regions to compute the regional context. Intuitively, the boundaries of the object regions provide the spatial relationship between the objects. In the same object region, the pixels contain the consistent category information. We resort to these nice properties of the object regions and construct the regional context, which enhances the pixel representations and eventually improves the segmentation performance.

---

***R1-Q1****: The authors should clarify the motivation of the step-by-step **RCB** and **RIB**.*

**A**: Thanks. The step-by-step **RCB** and **RIB** are motivated by the need for using the regional context to enhance the pixels. We use **RCB** to group the pixels of the image reasonably into different object regions. Based on the boundary and the representative pixels of the region, we construct the spatial and category representations of the object. Next, **RIB** exchange information between object regions, forming the regional context for enhancing the pixels.

---

***R1-Q2****: The authors should explain the extra structure after the backbone in contrast experiments.*

**A**: In Table 1-3 and 5, we use the backbone ResNet-101 without any extra structure (e.g., SPP). We equip the backbone with **RCB** and **RIB** to produce the contextual representation, which is fed to a $1 \times 1$ convolution layer and a softmax layer for segmentation. In Table 4, we use the backbone HRNetV2-W48 equipped with an ASPP structure (see "HRNetV2-W48+ASPP"), along with our approach, to make a fair comparison with the latest OCRNet [33].

---

***R1-Q3****: The authors should optimize the formula.*

**A**: Thanks. We will optimize the typesetting and reduce the notations of accumulation.

---

***R2-Q1****: Why the RANet capture more context information? The category information in RANet may be not accurate.*

**A**: In contrast to the SPP or attentional models that capture the pixel-level context, RANet captures the regional context. RANet allows the regional information to be exchanged between the pixels (see **RIB** in Figure 4), forming the pixel representations that contain the pixel-level and regional context. We agree that the category information may be inaccurate. Thus, we use **RCB** to select the representative pixels in different regions, based on the category and boundary information. Our approach produces more reliable context, compared to using the category information alone.

---

***R2-Q2****: How to compute the final segmentation map based on the contextual feature map O?*

**A**: The contextual map $O$ is fed to a $1 \times 1$ convolution layer and a softmax layer, for computing the segmentation map.

---

***R2-Q3****: What is the direction of the line in Eq.(1)?*

**A**: We illustrate an oblique line in Figure 2. Actually, the direction of the line is determined by the locations of the end pixels. Thus, the line can be vertical, horizontal or oblique.

---

***R3-Q1****: RCB and RIB both seem to be very time-consuming.*

**A**: Please note that **RCB** computes the semantic similarity based on the low-dimensional category score vectors. Though **RIB** needs to select the representative pixels, it effectively reduces the number of pixels that exchange context, and consequently saves the computation. In Table 3, we have shown that our approach can be done at the cost of the reasonable computation, compared to the latest attentional models.

---

***R3-Q2****: The authors should explain the differences between RANet and SPP or other attention mechanisms.*

**A**: Thanks for your valuable suggestion. Please see "**Motivation of our approach**".

---

***R3-Q3****: Why not use all the pixels in each region? The author needs to give a short explanation.*

**A**: Thanks. Using many pixels may involve the ambiguous information of the pixels that are near the object boundaries. It degrades the performance (see Table 1). Besides, using the representative pixels saves computation (see Table 3).

---

***R3-Q4****: Why do the representative pixels gradually separate from each other?*

**A**: We conjecture that the network optimization leads to the separation of the representative pixels to comprehensively represent different contents of the object. We will provide more analyses on the representative pixels in our future work.

[Meta-Review · NeurIPS 2020]

All the reviewers upgraded their review scores from “borderline accept” to “accept” during the discussion period to recognize the novelty and effectiveness of the proposed region aggregation contextual information aggregation and propagation scheme. The AC agrees with the view and would like to recommend acceptance of the paper.